# Shift of subtropical transport barriers explains observed hemispheric asymmetry of decadal trends of age of air

Gabriele P. Stiller[1], Federico Fierli[2], Felix Ploeger[3], Chiara Cagnazzo[2], Bernd Funke[4], Florian J. Haenel[1], Thomas Reddmann[1], Martin Riese[3], and Thomas von Clarmann[1]

[1]Karlsruhe Institute of Technology, IMK, P.O.B. 3640, 76021 Karlsruhe, Germany
[2]National Research Council, Institute for Atmospheric Sciences and Climate, Via Fosso del Cavaliere 100, 00199 Rome, Italy
[3]Institute for Energy and Climate Research - Stratosphere (IEK-7), Forschungszentrum Jülich, 52425 Jülich, Germany
[4]Instituto de Astrofísica de Andalucía (CSIC), Glorieta de la Astronomía s/n, 18008 Granada, Spain

*Correspondence to:* Gabriele P. Stiller (gabriele.stiller@kit.edu)

**Abstract.** In response to global warming the Brewer-Dobson circulation in the stratosphere is expected to accelerate and the mean transport time of air along this circulation to decrease. This would imply a negative stratospheric age of air trend, i.e. an air parcel would need less time to travel from the tropopause to any point in the stratosphere. Age of air as inferred from tracer observations, however, shows zero to positive trends in the Northern midlatitude stratosphere and zonally asymmetric patterns. Using satellite observations and model calculations we show that the observed latitudinal and vertical patterns of the decadal changes of age of air in the lower to middle stratosphere during 2002–2012 are predominantly caused by a southward shift of the circulation pattern by about 5 degrees. After correction for this shift, the observations reveal a hemispherically almost symmetric decrease of age of air in the lower to middle stratosphere up to 800 K of up to -0.25 years over the 2002-2012 period with strongest decrease in the Northern tropics. This net change is consistent with long-term trends from model predictions.

## 1 Introduction

An acceleration of the Brewer-Dobson circulation as a consequence of global warming has been predicted by climate models (Austin et al., 2007; Austin and Li, 2006; Garcia and Randel, 2008; McLandress and Shepherd, 2009; Oman et al., 2009; SPARC CCMVal, 2010). The mean age of stratospheric air (AoA), used as a diagnostic of the strength of the Brewer-Dobson circulation (Waugh and Hall, 2002), is expected to decrease since an air parcel would need less time from the stratospheric entry point at the tropical tropopause to its actual location (Waugh, 2009). Balloon-borne inert tracer measurements ($CO_2$ and $SF_6$), however, yielded zero to positive stratospheric mean age of air (AoA) trends in the Northern midlatitudinal stratosphere for 1975-2005 (Engel et al., 2009), which seemed to be in conflict with this expectation. A global analysis of AoA trends derived from MIPAS satellite data for 2002 to 2012 supported these results for Northern midlatitudes but provided, beyond this, a global picture which resisted any easy explanation (Stiller et al., 2012; Haenel et al., 2015). Negative trends of mean AoA were found for large parts of the Southern middle stratosphere and the tropics, revealing a dipole-like structure of mean AoA trends with opposite signs in the two hemispheres. Similar hemispheric asymmetries of trends of stratospheric trace species were derived from other observational data sets by various groups, corroborating the results from the MIPAS AoA analysis,

e.g. for ozone (Eckert et al., 2014; Gebhardt et al., 2014; Nedoluha et al., 2015a, b; Pawson et al., 2014), hydrogen fluoride (Harrison et al., 2016), hydrogen chloride (Mahieu et al., 2014), (H)CFCs (Chirkov et al., 2016; Kellmann et al., 2012), nitrous oxide (Nedoluha et al., 2015a), and carbonyl sulfide (Glatthor et al., 2016).

Ploeger et al. (2015b) performed an analysis of the time series of AoA as derived from the MIPAS $SF_6$ observations and the AoA derived from a Chemical Lagrangian Model of the Stratosphere (CLaMS) model run with a clock tracer included. For both data sets, the AoA changes from 2002 to 2012 in the lowermost stratosphere below 380 K were found to be negative equatorwards of 40° in both hemispheres, but slightly positive polewards of 40° and between 380 and 420 K for the lower latitudes (Ploeger et al., 2015b). This difference between trends in low and high latitudes hints at a decoupling of the deep and the shallow branch of the Brewer-Dobson circulation, consistent with the general picture presented in Birner and Bönisch (2011); Bönisch et al. (2011). Alternative explanations point at the importance of mixing processes (Ray et al., 2010; Garny et al., 2014; Ploeger et al., 2015b). Neither of these hypotheses, however, can explain the partly opposed trends in both hemispheres as inferred from satellite measurements of $SF_6$ (Stiller et al., 2012; Haenel et al., 2015).

This paper is organised as follows: After a description of data and methods used (Section 2), we determine the positions of the subtropical transport barriers and their change over time (Section 3). We apply the derived shifts to tracer distributions (Section 4) and AoA distributions (Section 5) and demonstrate that the observed temporal changes can be emulated by the southward shift of the circulation pattern. In Section 6 we discuss our findings and their implications in the context of other work. In the conclusions (Section 7) we summarise our main findings and discuss their implications for other approaches to determine a change in the Brewer-Dobson circulation.

## 2  Data and methods

We have used satellite data from MIPAS and MLS, CLaMS model simulations and ERA-Interim reanalysis data in our work.

### 2.1  MIPAS data

The Michelson Interferometer for Passive Atmospheric Sounding (MIPAS) on the Environmental Satellite (Envisat) (Fischer et al., 2008) has provided composition measurements of the upper troposphere to the mesosphere for the period of July 2002 to April 2012 (with a data gap from April to December 2004). The coverage is global, for day and night conditions, with more than 1000 profiles per day. Here we use AoA data derived from global stratospheric $SF_6$ distributions. Monthly zonal mean $SF_6$ distributions were converted into AoA distributions from which decadal age trends depending on altitude and latitude have been inferred (Stiller et al., 2012). A revised set of AoA trends based on improved $SF_6$ retrievals (Haenel et al., 2015) and covering the full measurement period of 2002 to 2012 is used here. The monthly zonal mean AoA data used within this study have a very good precision in the order of better than 1% (expressed and the standard error of the mean of the monthly zonal mean data) because of the large number (several hundred to more than thousand) of individual profiles averaged. They are potentially affected by latitude- and altitude-dependent biases (see Stiller et al., 2008), which however are of minor relevance here, because we are interested in the temporal changes for fixed altitude-/latitude bins, and not in the absolute values. $N_2O$

global distributions from MIPAS for 2002 to 2012 have been used to derive the positions of the subtropical transport barriers. The original data represented on a geometric altitude grid have been interpolated to eight potential temperature levels between 520 and 1100 K. The retrieval of $N_2O$ from MIPAS observations is described by Plieninger et al. (2015), while validation of the $N_2O$ data set is presented by Plieninger et al. (2016). For monthly zonal means of $N_2O$, the standard error is extremely small and irrelevant for our application. MIPAS $N_2O$ has been found to have a latitude- and altitude dependent bias of less than 15 ppbv (Plieninger et al., 2016). Again, as we are not interested in absolute values, since the determination of the transport barrier position do not depend on those, this is of minor relevance in the context here.

## 2.2 MLS data

$N_2O$ global distributions for 2004 to 2014 from Aura/MLS have been used to derive the position of the subtropical transport barriers. MLS data version 3.30 has been retrieved from the Jet Propulsion Laboratory data server and interpolated to eight potential temperature levels between 520 and 1100 K. The data set covers the full globe nearly from pole to pole with more than 3000 profiles per day. The retrieval and validation of MLS $N_2O$ data is described in Lambert et al. (2007). The precision of single MLS $N_2O$ profiles is assessed to be 7–38%, which, however, is irrelevant because we use monthly zonal means for which the uncertainties are negligible. A bias, if any, should be not larger than 5% and is, as discussed above, of no relevance for this study.

## 2.3 Model calculations

The model simulations presented in this paper have been carried out with the Chemical Lagrangian Model of the Stratosphere CLaMS (McKenna et al., 2002). The advective part of transport in CLaMS is based on three-dimensional forward trajectories, driven by ERA-Interim reanalysis winds. As CLaMS uses potential temperature as vertical coordinate above $\sigma = 0.3$ (where $\sigma = p/p_s$ is the orography following vertical coordinate near the surface, with $p$ pressure and $p_s$ surface pressure), the vertical transport in the model is driven by the total diabatic heating rate, taken from the reanalysis forecast. In addition, a parameter-isation of small-scale atmospheric mixing is included in CLaMS, depending on the deformation rate of the large-scale flow. The model resolution of the CLaMS simulations considered here is about 100 km in the horizontal direction. In the vertical direction, the resolution is about 400 m around the tropical tropopause (see Pommrich et al. (2014) for details). The model AoA used here has been calculated from a 'clock-tracer', an inert tracer with a linearly increasing source in the orography-following lowest model layer (boundary layer). Further details about the model set-up used here are given by Pommrich et al. (2014).

## 2.4 Trace gas gradient genesis

Miyazaki and Iwasaki (2008) presented a method to separate the tendency of latitudinal trace gas gradients along an isentrope into the effects of mean residual circulation and eddy transport. Mean transport may cause a sharpening of gradients related to horizontal 'stretching' and vertical 'shearing', or a shift of gradients by meridional and vertical advection. Eddy transport may sharpen gradients at the edges of strong mixing regions (e.g., the surf zone) due to the 'stair-step' effect or may dilute gradients

due to the 'smoothing' effect. We performed this separation for CLaMS simulated $N_2O$ fields and analysed all relevant terms. In our paper, we make use of the trace gas gradient, resulting from the gradient genesis terms and representing well their latitudinal positions. A detailed description of the method is provided in Appendix A2 where we also demonstrate that the $N_2O$ tracer gradient itself is a good representative, in terms of latitudinal position, for the various gradient genesis terms assessed by Miyazaki and Iwasaki (2008).

## 3 The temporal evolution of the positions of the subtropical transport barriers

For deriving the positions of the subtropical transport barriers, we follow the method described by Sparling (2000). Using this approach, Palazzi et al. (2011) analyzed the probability density functions (PDFs) of long-lived tracer observations from four different satellite instruments over the years 1992 to 2009. The minimum of the tracer PDFs identifies a region where transport is inhibited and is hence used to diagnose the latitudinal positions of the barriers. As opposed to latitudinal gradients, the position of the PDF minima do not depend on the particular tracer used and are immune against biases between different data bases. Palazzi et al. (2011) found a strong dependence of the transport barrier positions on the seasonal cycle and a significant impact of the phase of the quasi-biennial oscillation (QBO). A more detailed description of the method is provided in Appendix A1. We apply their method to a decadal analysis of the subtropical transport barrier positions by including longer periods of observational data and model simulations: we use $N_2O$ distributions from MIPAS (2002 to 2012), MLS (2004 to 2014), and from the Lagrangian transport model CLaMS driven by ERA-Interim reanalyses (1992 to 2014) (Fig. 1).

We determined the shift of the latitudinal positions of the transport barriers for eight levels of potential temperatures between 500 and 1000 K (appr. 19–36 km) by comparing the mean latitudinal positions of two time intervals: 2005–2008 (period I in the following) versus 2009–2012 (period II) (Fig. 2). The two periods were selected because data coverage was best for them, the QBO was in about the same phase at the beginning of the period, and a change in the barrier positions is obvious between the two periods. For both observational data sets and the CLaMS simulations a consistent picture is obtained: below 800 K, the Northern subtropical transport barrier moved towards the South by about -3° from period I to period II, while above 800 K, the emerging shifts are northward. The Southern subtropical transport barrier, however, moved southward for all potential temperature levels, with shift values increasingly negative with increasing altitude. This results in a southward shift and small narrowing of the tropical pipe below 800 K and a significant widening of the tropical pipe above. After 2012, however, MLS and CLaMS data indicate that the barrier positions below 800 K move northwards again, characterising the observation as variability on the time scale of few years.

In order to demonstrate the consistency of the transport barrier positions determined by us with well-known dynamical mechanisms discussed earlier in literature (Shuckburgh et al., 2001; Palazzi et al., 2011; Miyazaki and Iwasaki, 2008), we compare the transport barrier positions derived above with the positions of the maximum tracer gradient regions according to Miyazaki and Iwasaki (2008), with the autumn/spring positions of the turnaround latitudes ($\bar{w}^* = 0$) derived from ERA-Interim reanalysis data (see Fig. 3) and with zonal mean zonal winds. In general, all diagnostics for the transport barrier positions exhibit a large interannual variability, related to the QBO phase (visible in Fig. 3 as positive areas of zonal mean zonal winds).

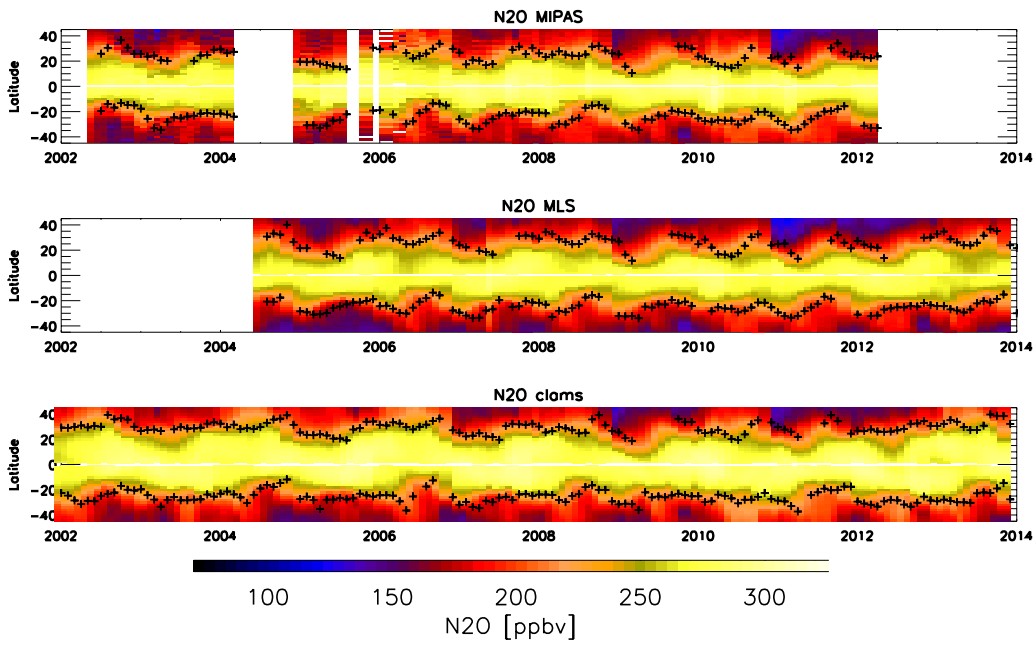

**Figure 1.** Time series of zonal N$_2$O distributions at the 600 K potential temperature level from MIPAS observations (top panel), MLS observations (middle panel), and CLaMS model simulations driven by ERA-Interim reanalyses (bottom panel). The black crosses indicate the positions of the subtropical transport barriers derived with the method by Palazzi et al. (2011). Gaps in the time series of transport barrier positions appear where the method for determining the positions failed.

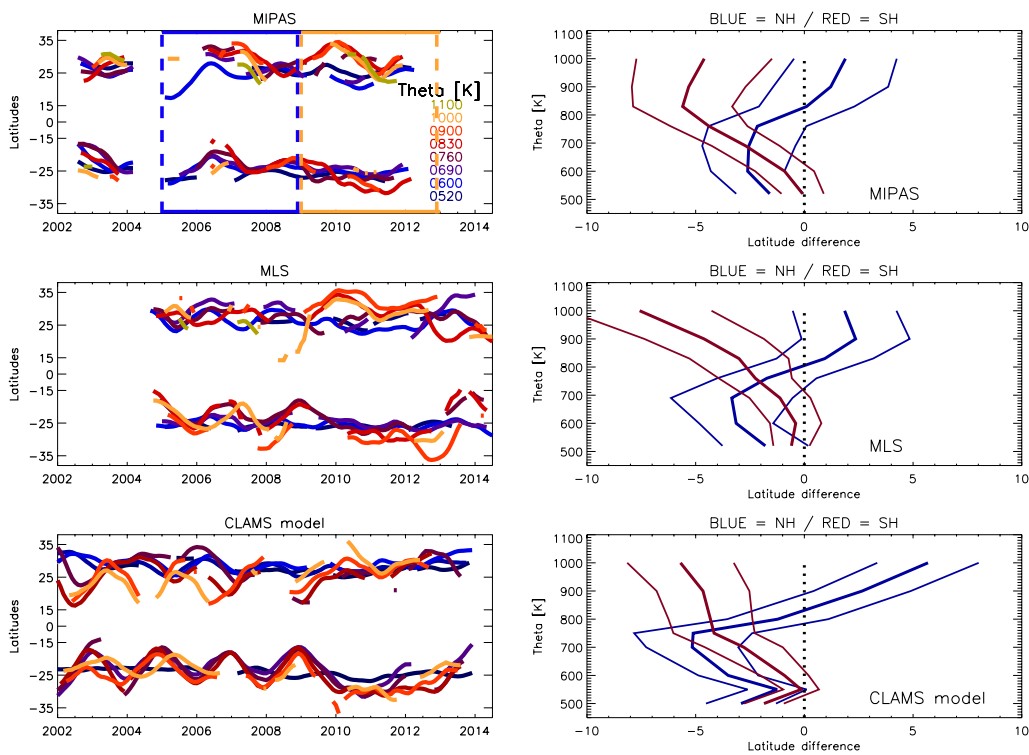

**Figure 2.** Positions of the subtropical transport barriers over time for eight potential temperature levels between 520 und 1100 K (colour coding see legend) from $N_2O$ monthly zonal mean distributions observed by MIPAS (top left), MLS (middle left), and simulated by CLaMS (bottom left). The right panels provide the vertical profiles of the shifts of the transport barrier positions (blue: NH; red: SH) determined as the difference of the mean positions of the 2005 – 2008 versus 2009 – 2012 period (vertical dashed lines in the top left panel) from the respective instrument on the left. The bold solid line is the shift profile, and the thin solid lines indicate the 1-$\sigma$ uncertainties of the shift values. The profiles have been smoothed by a running mean over three adjacent data points.

The shifts of both (NH and SH) barrier positions appear to be related strongly to the QBO phase, at least until 2009, exhibiting a northward shift of both barriers at the end of the positive QBO phase. With this finding we are consistent to Shuckburgh et al. (2001) and Palazzi et al. (2011) who have identified this coupling of transport barrier positions with the QBO on basis of simulations and HALOE observations, respectively. Besides the interannual variability, the positions of the transport barriers have a clear seasonal cycle, related to the gradient genesis terms reported in Fig. 3.

The turnaround latitudes ($\bar{w}^* = 0$) may also provide information on the position of the subtropical transport barriers as these separate the upwelling motion in the tropical pipe from the downward transport in the surf zones; however, the latitudinal positions of turnaround latitudes are not expected to be always identical with those of the transport barriers (Seviour et al., 2011). For the Northern hemisphere, the positions of the turnaround latitudes (transfer from blue to red colours in Fig. 3) for autumn and spring are in good agreement with the positions of the subtropical transport barriers derived from satellite tracer observations, while for the Southern hemisphere, neither the positions nor the shift agree well. At the intra-seasonal timescales, the turnaround latitude positions are related to the shearing and vertical advection terms of the tracer gradient genesis that are represented by the maximum tracer gradient regions in Fig. 3 (black contours), according to Miyazaki and Iwasaki (2008) (for more details, see Appendix A2). The maximum tracer gradients are in turn correlated with minimum PDFs. Indeed, the positions of the gradient genesis regions and their shifts agree well with those derived from satellite tracer observations.

All three quantities indicating the positions of the subtropical transport barriers show a rather sudden shift towards the South around the year 2009 ($\bar{w}^*$ only for the Northern hemisphere). This shift is most clearly reflected in the locations of the areas of maximum meridional $N_2O$ gradients (black contour lines). The analyses of the location of the gradient genesis and the $\bar{w}^*$ turnaround latitudes demonstrate that changes in the circulation are responsible for the shifts found in the tracer abundance fields observed by MLS and MIPAS, and that this process is also contained in ERA-Interim reanalysis data.

Figure 4 compares the mean positions of the transport barriers for period I and II, respectively, derived from the three different metrics for the 600 K potential temperature levels. The barrier positions from the tracer PDF analyses of MIPAS (triangles), MLS (circles) and CLaMS (vertical lines) agree very well, in particular for the Southern hemisphere. For the Northern hemisphere, all three metrics find a considerable and consistent southward shift while for the Southern hemisphere, the shift is very small, consistent to Fig. 2. The positions of the extrema of the tracer gradients (minimum for NH, maximum for SH) agree well with the PDF-based metrics in the Northern hemisphere, and a shift to the South between the periods can be identified as well. In the Southern hemisphere, the positions of the maximum tracer gradients are found further North than the PDF-based positions, and there is no significant shift of the positions between the two periods. The $\bar{w}^*$ turnaround latitudes are roughly at the same position as the other metrics in the Northern hemisphere, but far more southward in the Southern hemisphere; for the Northern hemisphere, a shift towards the South from period I to period II is obvious as well.

## 4   Shift of the global circulation pattern

The meridional gradient determining the subtropical transport barrier is generated by latitudinal variations in $\bar{w}^*$, by the strength of the mixing and wind shear (Miyazaki and Iwasaki, 2008). Hence a shift in the position of the PDF minima (i.e. minimum

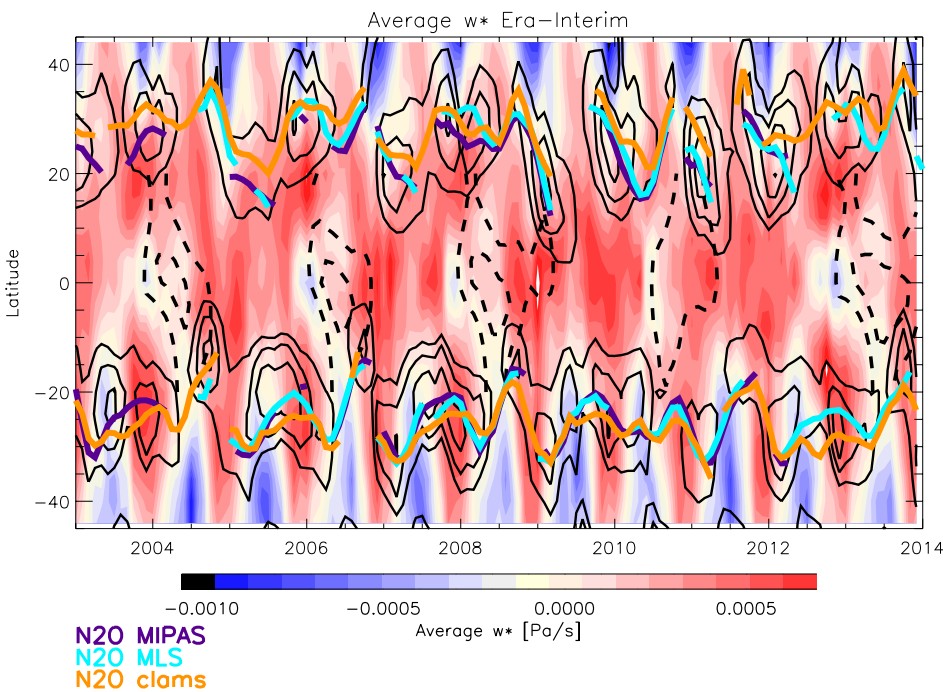

**Figure 3.** Positions of the subtropical transport barriers over time at 600K compared to other dynamical quantities providing a measure for the position of the tropical pipe: the purple, turquoise and orange solid lines provide the positions of the subtropical transport barriers from MIPAS, MLS, and CLaMS data, respectively, derived with the method of Palazzi et al. (2011); the color coding indicates the values of $\bar{w}^*$ (see color bar); the latitude regions of maximum meridional tracer gradients are indicated as black contour lines. Black dashed lines enclose the regions of positive zonal-mean zonal wind (QBO positive phases).

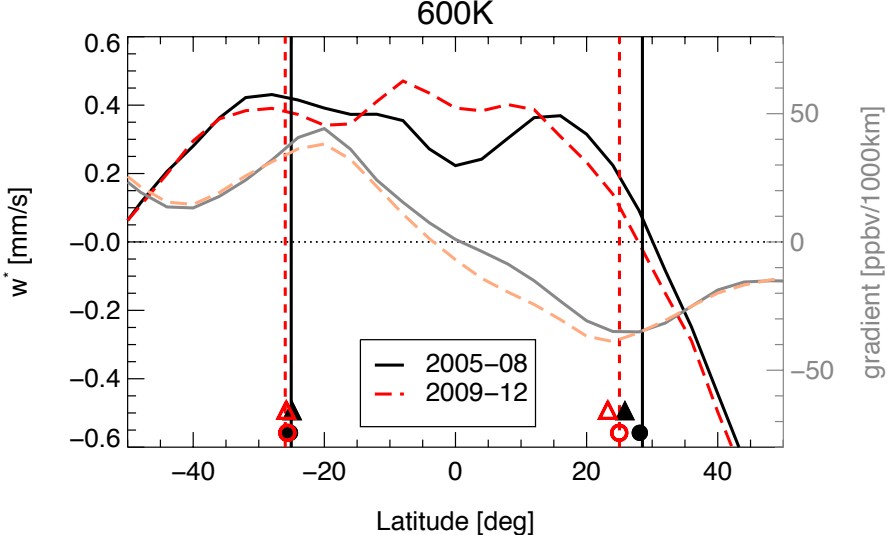

**Figure 4.** Latitudinal positions of the transport barriers for the two periods I and II at the 600 K level; black/grey colours refer to period I, red/orange colours to period II. Mean $\bar{w}^*$ as function of latitude for the two periods is provided as solid black and dashed red line (left axis); the meridional tracer gradients, representing the gradient genesis terms after Miyazaki and Iwasaki (2008) for the two periods are provided as grey and dashed orange lines (right axis). The vertical lines indicate the latitudinal positions of the transport barriers derived from the tracer PDF analysis of CLaMS data, while circles provide the barrier latitudes from the tracer PDF analysis of MLS $N_2O$ data, and triangles the barrier latitudes from the tracer PDF analysis of MIPAS $N_2O$ data).

transport) may be related to a shift in the entire circulation pattern. If the shift of the tropical pipe was indicative of a shift of the complete circulation pattern in this altitude regime, we would expect, at first order, the zonal distribution pattern of any long-lived tracer to be shifted by the same amount. This is because the zonal mean distribution of a long-lived tracer in the stratosphere is predominantly determined by the residual circulation and the additional eddy mixing. We therefore used

the CLaMS $N_2O$ zonal mean distribution for the first period of interest (period I, 2005–2008), and applied altitude-dependent latitudinal shifts according to the values derived for the transport barrier positions. This was done for all latitudes polewards of 10°N/S. At tropical latitudes appropriate shift values have been calculated by linearly interpolating from the derived values at 10°N/S, and the distributions have been accordingly shifted. The shift effect to the distributions has finally been calculated by subtracting the unshifted original distribution from the shifted distribution. The altitude/latitude pattern of these differences

is then compared to the altitude/latitude pattern of $N_2O$ decadal changes as modelled with CLaMS (Fig. 5). Equatorwards of about 60° N/S, the pattern of the CLaMS-modelled decadal changes of the zonal mean distribution of $N_2O$ is explained to a large degree by the shift of the $N_2O$ zonal mean distribution. Overall, the remarkable agreement of the patterns indicates that a shift of the entire circulation pattern is largely responsible for the changes in the tracer distribution. Remaining differences in the change versus shift patterns hint towards other competing processes taking place as well. The absolute amounts of the

changes are larger than those derived by shifting the distribution, hinting also towards such competing processes.

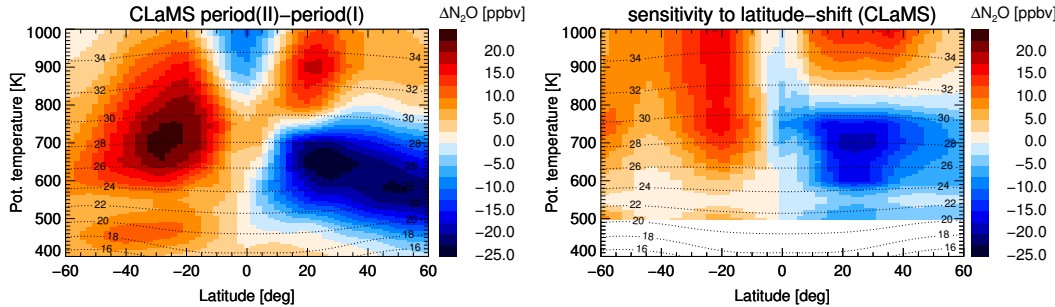

**Figure 5.** Difference of CLaMS zonal mean $N_2O$ distributions between the periods I and II (left), compared to the effect of a shift of the $N_2O$ zonal mean distribution of period I according to the altitude dependent latitude shifts of the subtropical transport barriers as derived for CLaMS (Fig. 2, top left panel, orange lines) (right).

Similar hemispheric dipole patterns in linear trends of stratospheric tracers, or in differences of tracer distributions between two periods, have been observed for a number of trace species from several observations: Mahieu et al. (2014) found an increase in HCl volume mixing ratios in the Northern hemisphere and a decrease volume mixing ratios in the Southern hemisphere between 2005/6 and 2010/11 for 10–25 km altitude and related this to a respective change in AoA derived from

SLIMCAT and KASIMA model simulations. Harrison et al. (2016) detected a similar dipole pattern in the change of HF between 2004 and 2012. Kellmann et al. (2012) found CFC-11 and CFC-12 changes over the period of 2002 to 2012 that were smaller than the trends at the surface for the Northern hemisphere and larger than the surface trends in the Southern hemisphere. Similar deviations from surface trends were found for HCFC-22 for the period 2005–2012 (Chirkov et al., 2016). Nedoluha et al. (2015a) and Nedoluha et al. (2015b) found hemispheric asymmetries and dipole patterns in trends of $N_2O$

and ozone for the period 2004–2013 by analyzing Aura/MLS data. Dipole patterns in ozone trends were also seen in ozone data from other satellite instruments (MIPAS, SCIAMACHY) for 2002–2012 (Eckert et al., 2014; Gebhardt et al., 2014). The signs of all these observed changes are consistent with a southward shift of the tracer distributions. Eckert et al. (2014) even demonstrated in their paper (their Fig. 19) that a shift of the ozone distribution at the beginning of their time series by about 5 degrees to the South could explain the observed 10-year changes to a large degree. It is therefore very likely that all these

observations can be explained by the southward shift of the circulation pattern as well.

## 5   Effect of the shift of the stratospheric circulation pattern on observed and modelled mean age of air

The shift of the circulation pattern to the South below 800 K and the widening of the tropical pipe above should be detectable also in the AoA distribution in the stratosphere. We apply the latitudinal shift as detected from the observed and modelled

subtropical transport barrier positions to the global zonally averaged AoA distribution of period I derived from MIPAS observations and modelled by CLaMS.

For MIPAS, the dipole-like structure in the AoA decadal change in the middle stratosphere between 500 and 800 K is well reproduced by the effect of the shift of the transport barrier positions, applied to the distribution of period I (Fig. 6, top right panel). However, there remain differences in the upper stratosphere above 800 K and in the Southern hemisphere southward of 40° S. Both regions are strongly affected by advection of $SF_6$-depleted mesospheric air. Since the amount of $SF_6$

depletion in the mesosphere is proportional to the available $SF_6$ abundance, the increase of $SF_6$ in the atmosphere itself leads to an increasing $SF_6$ loss which in turn appears as a non-real additional positive AoA trend (Stiller et al., 2012; Haenel et al., 2015). This artefact may outweigh the negative change of AoA. Another possible explanation for this observed discrepancy is that the widening of the tropical belt above 800 K does not affect the mid-latitudes in a similar way than the shift of the subtropical transport barriers at lower altitudes. Further, the observed change is negative in the Northern tropics and the dipole

is asymmetric to the equator, while the AoA changes produced by the shift of the distributions are symmetric to the equator, as expected from the symmetric AoA distribution itself.

For CLaMS AoA trends, the situation is clearer, and the model simulations confirm the findings from the observations. Below 800 K, the pattern of the AoA change between period I and period II (Fig. 6, bottom left panel) has a similar dipole structure as the MIPAS observations, with widely negative trends in the Southern hemisphere, and positive trends in the Northern

middle stratosphere (Ploeger et al., 2015b). In contrast to the MIPAS data, we find negative trends in the Northern upper stratosphere; this is because CLaMS uses a perfect clock tracer for AoA, that is not affected by chemical loss as $SF_6$. The shift of the circulation pattern to the South reproduces the pattern of the trends extremely well (Fig. 6, bottom right panel). However, similar to $N_2O$ (Fig. 5), the changes due to the shift are weaker than the modelled changes, hinting towards additional competing processes. The negative change in the Northern hemisphere above 800 K is explained by the widening of the tropical pipe there

(see Fig. 2, bottom right panel). In contrast to the MIPAS observations, the CLaMS changes are positive in the Northern tropics below 800 K.

The good general agreement between MIPAS and CLaMS confirms that the derived shift of the tropical pipe and the surf zones is represented as well in the ERA-Interim meteorological data that are used to drive CLaMS. However, observed differences like the negative change in the Northern tropics hint to competing processes which cannot be explained by the shift of

the circulation pattern alone.

## 6   Discussion

We have shown that the observed dipole pattern in changes of stratospheric AoA during the last decade (2002–2012) (Stiller et al., 2012; Haenel et al., 2015), with negative changes in the Southern hemisphere and positive changes in the Northern hemisphere can be explained to a large part by a shift of the stratospheric circulation pattern. Both MLS and MIPAS satellite

observations and the ERA-Interim reanalysis show clear evidence for this shift. Indications for the shift are the changes of the positions of the subtropical transport barriers. The transport barriers have been shown to move southwards below 800 K in both hemispheres over the period 2005 to 2012 by about 5°. Above 800 K, the transport barriers diverge, leading to a substantial widening of the tropical pipe at these higher levels. The positions of the transport barriers have been derived from

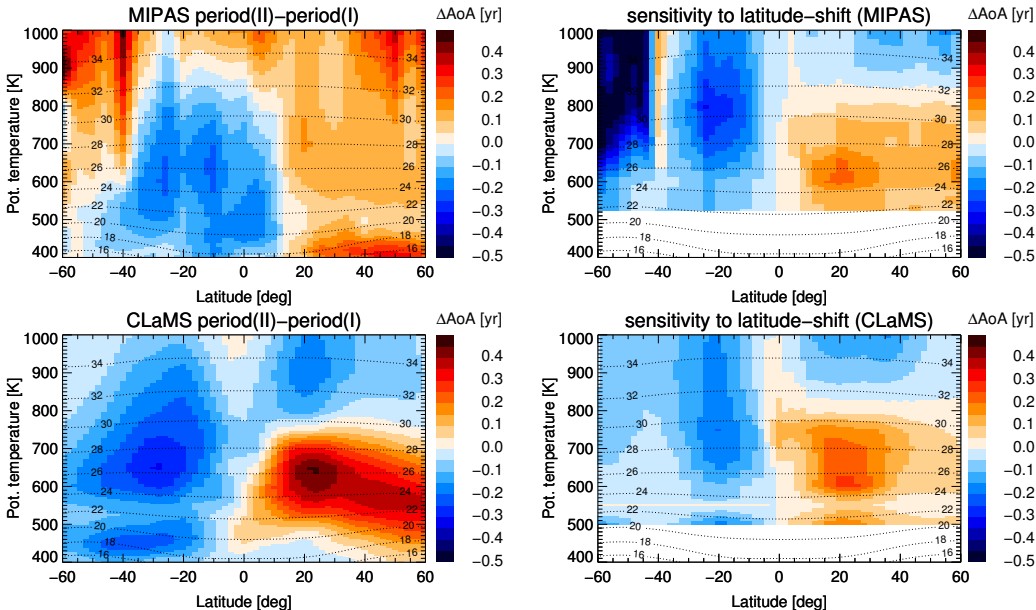

**Figure 6.** Top: Difference of MIPAS zonal mean AoA distributions between the periods I and II (left), compared to the effect of a shift of the zonal mean AoA distribution of period I according to the altitude dependent latitude shifts of the subtropical transport barriers as derived for MIPAS (Fig. 2, bottom right panel, blue lines) (right). Bottom: The same for CLaMS zonal mean AoA.

both observational and modelled tracer fields and provide a consistent picture, indicating that ERA-Interim meteorological data that drive the model calculations exhibits this shift of the circulation pattern as well. Moreover, comparison to direct diagnostics from ERA-Interim, the $\bar{w}^*$ turnaround latitude and the latitude of the meridional trace gas gradient genesis (Miyazaki and Iwasaki, 2008) demonstrate consistency of our findings regarding the shifts of the transport barrier positions with the general

dynamical mechanisms.

We cannot exclude additional influences on the stratospheric trace gas distributions, like potential effects of changes in the upward mass flux due to a changing volume of the upwelling region related to the southward shift, or of changes in the permeability in the subtropical transport barriers. Also, it is to be expected, that the age spectra change between period of strong and of reduced interannual variability, leading to differing mean ages of air. However, the good agreement between the

decadal change and the effect of shifting the global trace gas distributions provides strong evidence that the pure shift effect is the dominant cause for the observed AoA and trace gas changes during the last decade.

The positions and latitudinal extensions of the surf zones that are separated from the tropical pipe by the subtropical transport barriers are determined by the latitude where planetary waves can propagate into the stratosphere and break there. The propagation of planetary waves is hindered by the tropical easterly jet and, therefore, the positions of the subtropical mixing

barriers are closely related to the position of the jet. Hence, the observed southward shift might be related to changes in the position of the jet, similar to the findings regarding longer term trends in jet positions by Hardiman et al. (2013).

Decadal changes, as discussed here, may be largely influenced by natural variability. As a prominent mode of variability in the lower stratosphere, the Quasi-Biennial Oscillation (QBO) affects wave propagation and likely the position of the subtropical transport barriers, with planetary waves and associated mixing penetrating to lower latitudes during the westerly phase (Gray and III, 1999; Shuckburgh et al., 2001; Palazzi et al., 2011). A strong hint, that either changes in the QBO are reflected in the positions of the transport barriers, or both, QBO and the transport barrier positions, are affected by another yet unknown quantity, is the fact that at least in the CLaMS barrier position time series, the QBO variability changes dramatically between period I and period II (see Fig. 2, lower left panel). However, any cause-and-effect relationship cannot be derived from this correlation result.

Calvo et al. (2010) demonstrated that sea surface temperature anomalies related to strong El Niño and La Niña events also impact the propagation of planetary waves into the stratosphere and have the potential to shift the positions of the subtropical jets and the surf zones. A very recent model study of Garfinkel et al. (2016) shows that climate model simulations may include members with increasing mean age in the NH. Therefore a realistic representation of natural variability in model simulations is crucial if comparisons of decadal circulation trends with observations are made.

While the cause for the circulation shift is an important research issue in itself, our analysis provides evidence that the AoA observations by MIPAS do not generally contradict the theoretical expectation of an accelerated Brewer-Dobson circulation but that the latter effect can simply be masked by competing processes of possibly shorter time scales. Figure 7 shows the remaining AoA trend from MIPAS after subtracting the part which can be explained by the shift of the circulation pattern. After this correction the dipole pattern has disappeared and the change of age of air follows widely the idea of an accelerated deep branch of the Brewer-Dobson circulation below 800 K at tropical to mid-latitudes. In the mid-latitude middle stratosphere (40-50°N, 24-34 km), however, where Engel et al. (2009) found a zero to slightly positive AoA trend over the years 1975 to 2005, the shift-corrected AoA trend from MIPAS indeed remains positive. The negative trend could indicate either an acceleration of the BDC (accompanied by related changes of isentropic mixing, see e.g. Ploeger et al. (2015a)) or an additional upward shift of the circulation pattern as suggested by Oberländer-Hayn et al. (2016). In any case, the strongest negative trend of about -0.25 years/decade occurs in the Northern tropics and is consistent with trends derived from model calculations (e.g., Waugh, 2009). The strong positive changes above 800 K, however should be taken with caution, because, as explained earlier, an artefact due to in-mixing of mesospheric $SF_6$-depleted air, leading to artificial positive AoA trends, cannot be excluded.

## 7   Conclusions

We have demonstrated that the observed spatial pattern of the trend of stratospheric mean age of air from 2002 to 2012 with different signs on the two hemispheres can be explained, to a large part, by a southward shift of the stratospheric circulation pattern between the potential temperature levels of approximately 500 and 800 K. This hypothesis of a southward shift is corroborated by a simultaneous southward movement of the subtropical transport barriers over this period in the order of zero to five degrees, dependent on altitude. We have demonstrated this southward shift of the subtropical transport barriers by means of various measures of their latitudinal positions: first, we have derived the latitudinal positions from tracer distributions from

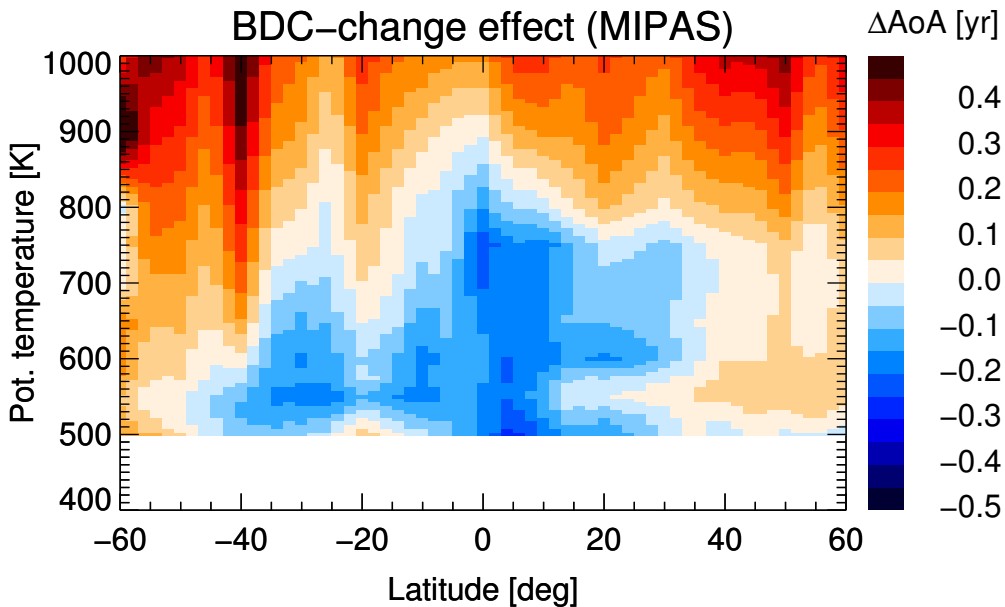

**Figure 7.** Difference between the observed AoA changes from MIPAS between period I and II and the effect of the derived shift of the circulation pattern. These remaining changes cannot be explained by the shift of the circulation pattern and are caused by competing processes.

independent satellite observations; then, in order to demonstrate the consistency of these positions with the general dynamical mechanisms, we have compared the positions to the turnaround latitudes of the residual circulation vertical velocity $\bar{w}^*$; and to the latitudinal positions of the regions of meridional trace gas gradient genesis following Miyazaki and Iwasaki (2008).

Further, we have shown that the chemistry-transport model CLaMS driven by ERA-Interim meteorology reproduces the observed southward shift of the transport barriers. This confirms that the ERA-Interim reanalysis data contain information on the dynamical processes leading to the observed shift. The modelled changes of zonal mean tracer distributions like $N_2O$ can largely be emulated by shifting the distributions to the South, which shows that not only the mixing barriers but the entire stratospheric circulation pattern has experienced this southward shift. The fact that the full hemispheric circulation, or at least the positions of the surf zones over their full latitudinal extensions, are coupled to the positions of the subtropical transport barriers is a new and interesting finding in itself.

Both the observed and modelled mean age of air trend patterns can be emulated by applying the same southward shift to the age of air zonal mean distribution. The age of air change in the two hemispheres is therefore to be understood as a local change: the age of air in a certain latitude band changes according to the shift of latitudinal position of the circulation; it does not directly hint towards an acceleration or deceleration of the Brewer-Dobson circulation or related isentropic mixing effects. Indeed, Ploeger et al. (2015a) have shown that the age of air changes shown in Fig. 6 (bottom left) are mainly due to a trend in aging by mixing; again, this trend is caused by a shift of the latitudinal regimes where aging by mixing takes place. In contrast

to this, the remaining observed age of air trend after correction for shift effects (Fig. 7) seems to indicate true changes in the residual circulation, either due to changes in the residual transport times or the isentropic mixing or both.

The identification of potential mechanisms responsible for the observed northward/southward movement of the subtropical transport barrier and surf zones is an interesting research topic on its own and beyond the scope of this paper. We have shown that such kind of processes acting on time scales of (less than) decades may have extensive impact on the distribution of tracers and the stratospheric age of air. They cannot be ignored when observational data are to be interpreted. This is particularly true if the analysis is fixed to certain latitude bands as it is often done to derive changes in the tropical uplift from tracer distributions like ozone. Ignoring a latitudinal shift of the upwelling region over time may lead to erroneous conclusions about the change of the strength of the Brewer-Dobson circulation. Vice versa, trends of age of air or tracers cannot fully be derived from observations within an oversimplified picture of a scalar and latitudinally fixed acceleration or slowing down of the Brewer-Dobson circulation. In particular, an adequate representation of natural variability in models appears as a necessary prerequisite, and it is important to take the natural variability both in the models and observational data into account when making comparisons, especially for trends over relatively short periods.

## 8  Data availability

MIPAS data are available from http://www.imk-asf.kit.edu/english/308.php (upon registration) and from the corresponding author (Gabriele.Stiller@kit.edu). MLS data are available from https://mirador.gsfc.nasa.gov (upon registration). The CLaMS model data may be requested from the corresponding author (f.ploeger@fz-juelich.de). ERA-Interim data were retrieved directly from the ECMWF web site http://www.ecmwf.int/en/research/climate-reanalysis/era-interim.

# Appendix A: Methods for determination of the latitudinal positions of the transport barriers

Three different metrics have been used to determine the latitudinal positions of the subtropical transport barriers: the subtropical minimum in the probability distribution function (PDF) of a long-lived tracer (Sparling, 2000; Neu et al., 2003; Palazzi et al., 2011); the gradient genesis area (Miyazaki and Iwasaki, 2008); and the turnaround latitude of the vertical component of the transformed Eulerian mean (TEM) residual circulation $\bar{w}^*$, i.e. the latitude where $\bar{w}^* = 0$ (e.g. Seviour et al., 2011). In the following we detail these metrics.

## A1   Probability distribution functions of the long-lived tracer $N_2O$

For each hemisphere, the subtropical minimum of the probability distribution function (PDF) $P(\chi)$ of the tracer $\chi$ occurs at a precise mixing ratio value of the tracer, $\chi^*$, the so-called "tracer boundary"' of the subtropics. The "support" of the subtropical tracer boundary can be defined as the region in latitude space over which the tracer field takes on values in a neighbourhood of $\chi^*$. From a mathematical point of view, this translates in the calculation of the PDF of the support of $\chi^*$, that is, the conditional PDF $P(\phi|\chi^*)$. $P(\phi|\chi^*)$ is the probability distribution of the latitudes of observations having mixing ratios near $\chi^*$. The most probable value of $P(\phi|\chi^*)$ identifies the latitudinal position of the subtropical mixing barrier, $\phi^*$.

The middle part of Fig. A1 shows the probability density distribution for finding a certain $N_2O$ certain volume mixing ratio at a certain latitude $\phi$, i.e. $P(\phi|\chi)$, for January 2010 for the Northern and the Southern hemisphere. On the left and right side of the top middle panel, the probability of finding a certain $N_2O$ certain volume mixing ratio $\chi$ integrated over all latitudes, i.e. $P(\chi)$, is shown. In the Northern hemisphere the PDFs have maxima at three latitude regions, namely the tropics, the surf zones, and the polar vortex region, and two minima, namely the subtropical and the vortex transport barrier. In the Southern hemisphere, the polar maximum and vortex barrier minimum is missing due to summer conditions.

In a first step, $\chi^*$ is searched as the minimum of the PDFs integrated over all latitudes; the horizontal double lines in Fig. A1 indicate this minimum and its uncertainty range for both the NH and the SH. In a second step the latitude $\phi^*$ where $P(\phi|\chi^*)$ is maximum is determined (lower panel in Fig. A1, with the vertical red solid and dotted lines indicating the maximum positions $\phi^*$ and their uncertainties, respectively). This is the most probable latitudinal position of the subtropical transport barriers. In cases were the minimum of the tracer probability $P(\chi)$ is very broad and shallow or not unique, the method fails to determine the barrier positions.

The correlation of the monthly positions of the mixing barriers derived from MIPAS and MLS $N_2O$ volume mixing ratios, respectively, is provided in Fig. A2, for potential temperature levels between 520 and 830 K and for four seasons. The vertical and horizontal extensions of the crosses indicate the uncertainties of the individual position estimates. The positions derived from the two data sets reveal an excellent correlation close to the 1:1 line. The PDFs of their differences are provided as solid curves, with maxima close to zero. Their standard deviations are always larger than the maximum values; this indicates that the difference from zero is not significant and the positions derived from the two data sets are identical within their uncertainties. The average uncertainties of the positions derived from MIPAS and MLS is in the order of 1.1 to 1.5 degrees.

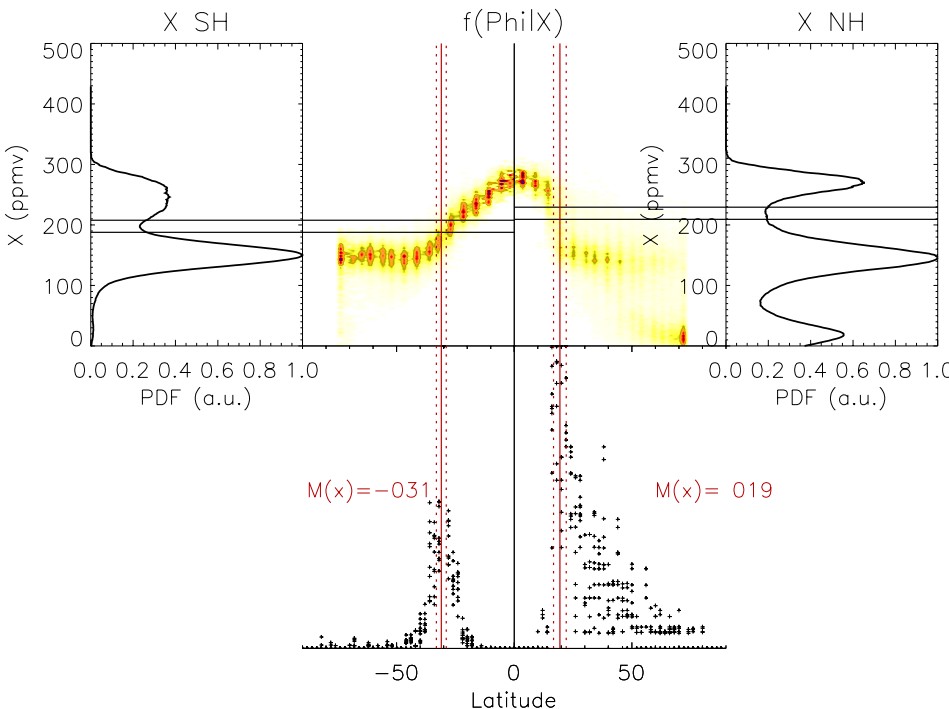

**Figure A1.** Schematic to illustrate the derivation of the transport barrier positions from tracer PDFs at the example of the $N_2O$ distribution from CLaMS for January 2010. The middle figure in the top row shows the PDF to find a certain $N_2O$ volume mixing ratio at a certain latitude (color coding, arbitrary units), i.e. $P(\phi|\chi)$, with latitude as horizontal and volume mixing ratio as vertical axis. On the left and right in the top row the PDFs to find a certain volume mixing ratio integrated over all latitudes, i.e. $P(\chi)$ for the Northern (right) and the Southern (left) hemisphere is shown. The horizontal double lines indicate the volume mixing ratio range for the minimum of $P(\chi)$ related to the subtropical mixing barriers, i.e. their values $\pm$ their uncertainties. In the lower panel, the latitude with the highest probability to find these mixing ratios is identified (indicated by the red vertical lines). The red dotted lines provide the uncertainty ranges determined by the uncertainties of the minimum vmrs. The red numbers indicate the latitudes of the barrier positions found.

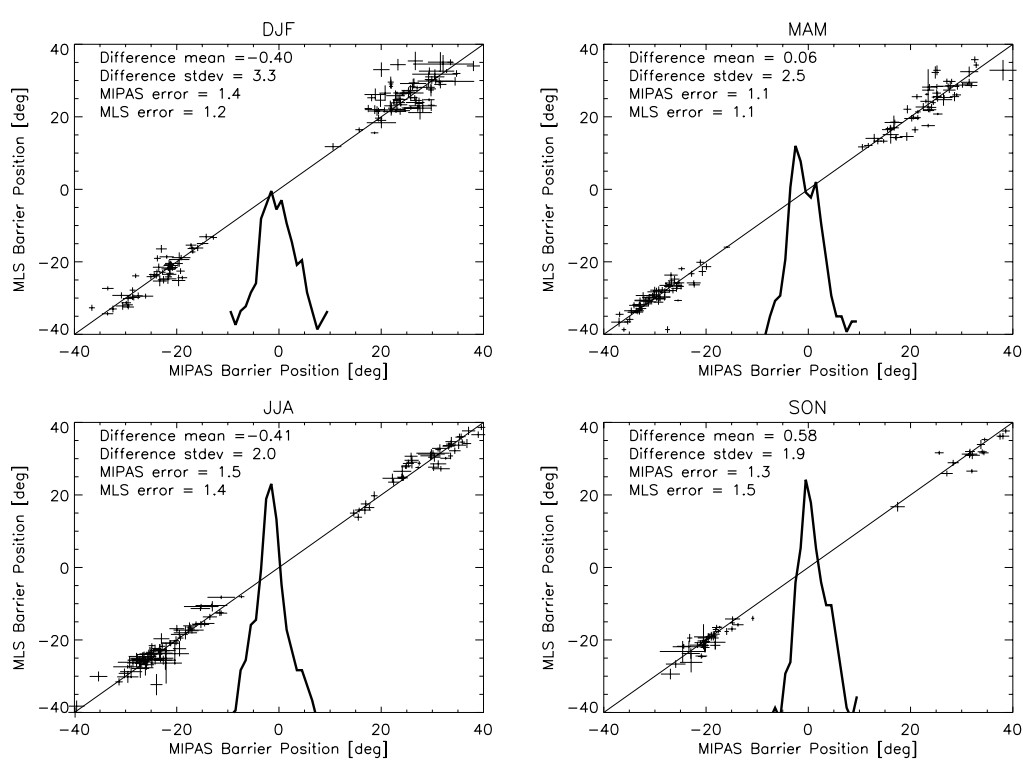

**Figure A2.** Correlation of latitudinal positions of the subtropical transport barriers (crosses) for the levels between 520 and 830 K, derived from MIPAS (horizontal axis) and MLS (vertical axis), respectively, for four seasons; the horizontal and vertical extensions of the crosses denote the 1-$\sigma$ uncertainties of the transport barrier positions from MIPAS and MLS, respectively. The solid curve is the PDF of the position differences between MLS and MIPAS. The legends provide the mean and the standard deviation of the differences of the barrier positions derived from the two data sets, and the average uncertainties of the positions from MIPAS and MLS data, respectively, derived from the support statistics (see Fig. A1).

## A2 Gradient genesis regions

Miyazaki and Iwasaki (2008) have formulated a gradient genesis equation using mass-weighted isentropic zonal means of $N_2O$ used as transport tracer. They have separated the tendency of the meridional gradient of the tracer into terms for the mean and the eddy transport (their equations (1) and (2)). We follow strictly this approach and calculate the terms provided by Miyazaki and Iwasaki (2008) as given in their equations (1) and (2) and shown in their Fig. 8. Figure A3 provides the transport and eddy terms similar to their Fig. 8 derived from CLaMS $N_2O$ fields and ERA-Interim reanalyses, except that we show in the bottom right panel the meridional tracer gradient only (and not the eddy vertical transport term). The seasonal variation of these terms and the positions of their minima and maxima reproduce well the findings from Fig. 8 of Miyazaki and Iwasaki (2008). The meridional tracer gradient resulting from the tracer genesis terms indicates very well the positions of the gradient genesis areas. For this reason we have shown the meridional gradient itself in Figs. 3 and 4. As demonstrated in Fig. A3, the absolute positions and seasonal evolution of the meridional tracer gradients agree very well with the positions of the transport barriers derived from the PDF analysis.

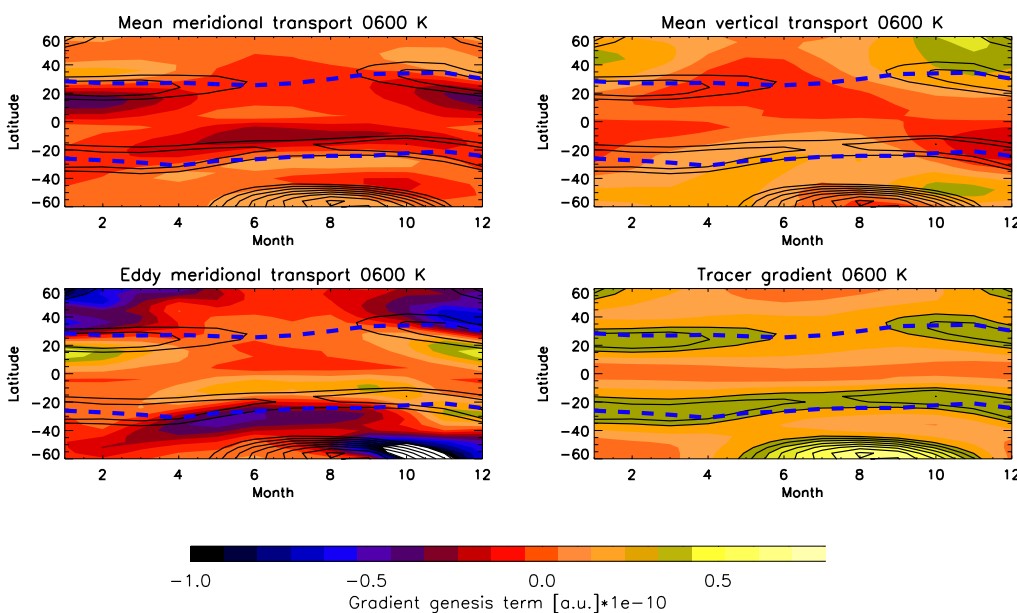

**Figure A3.** Seasonal variation at 600 K of the gradient genesis terms according to Fig. 8 of Miyazaki and Iwasaki (2008) from multi-annual means derived from CLaMS data and ERA-Interim reanalyses; the colour coding indicates the values of the gradient genesis terms; top left: mean meridional transport; top right: mean vertical transport; bottom left: eddy meridional transport; bottom right: tracer gradient. The black solid contour lines mark the multi-annual mean regions of maximum tracer gradients and are the same in all four panels. The blue dashed lines show the transport barrier positions derived from CLaMS $N_2O$ data.

## A3 Turnaround latitudes

Turnaround latitudes are calculated from ERA-Interim data as

$$\bar{w}^* = \bar{w} + \frac{1}{a cos\phi}\left(cos\phi \frac{\overline{\nu'\theta'}}{\overline{\theta_z}}\right)_\phi = \frac{1}{a\rho_0 cos\phi}\frac{\partial\psi}{\partial\phi} \tag{A1}$$

with $\nu$ and $w$ the meridional and vertical components of the velocity, $a$ the Earth's radius, $\phi$ the latitude, $z = \log(\text{pressure})$, $\rho_0 = exp(-z/H)$, $H$ the density scale height, and $\theta$ the potential temperature. The overbar represents the zonal mean, and the prime is the deviation from the mean. $\bar{w}^* = 0$ provides the turnaround latitude.

## Appendix B: Uncertainty of the shift of the transport barrier and its impact on the zonal mean trend patterns

The uncertainties of the mean latitudinal positions of the transport barriers for period I and II, respectively, are determined from the monthly positions for each potential temperature level as standard error of the mean (SEM). The SEM of the differences of mean positions for period I and II, respectively, are calculated from these by Gaussian error propagation. They are provided in Fig. 2 as 1-$\sigma$ error ranges.

The uncertainties of the shifts transfer to the uncertainties of the shifted AoA patterns. We have assessed this by shifting the zonal mean AoA distributions of period I from CLaMS with shift profiles constructed as $\mathbf{S} \pm \Delta\mathbf{S}$ ($\mathbf{S}$ is the shift profile, $\Delta\mathbf{S}$ is its uncertainty). The result of this assessment is presented in Fig. A4. As expected, the shifted pattern is very sensitive to the amount of the shift. For the case $\mathbf{S} + \Delta\mathbf{S}$ where the northern Hemisphere shift is closer to zero in the lower part (below 800 K) and larger positive in the upper part of the stratosphere, we find strongly reduced AoA changes below 800 K, and increase AoA changes above 800 K, compared to the shifts shown in Fig. 6, bottom right panel. For the $\mathbf{S} - \Delta\mathbf{S}$ case, the effect is strongly increased below 800 K. Interestingly, the resulting pattern resembles very much the changes of AoA from period I to period II as modelled by CLaMS, also in their quantitative values. This demonstrates that the changes modelled by CLaMS are within the uncertainty range of the derived shifts.

*Author contributions.* G.P.S. initiated the study, coordinated the various analyses and wrote the manuscript. F.P. set up and analysed CLaMS model simulations, prepared most of the figures, and contributed to the analysis of the results and the writing of the manuscript. F.F. derived the positions of the transport barriers from satellite data, analysed ERA-Interim data, and contributed to the analysis of the results and the writing of the manuscript. C.C. contributed to the analysis and interpretation of the ERA-Interim-derived diagnostics. B.F. initiated the study by discussions on the structure of trace gas trends, and contributed to all further discussions and the writing of the manuscript. F.H. provided MIPAS satellite data of stratospheric mean age of air. T.R. provided expert advice on age of air in model simulations. M.R. contributed to the analysis and interpretation of the results. T.v.C. provided expert advice with satellite data, and contributed to the analysis and interpretation of the results and to the paper writing.

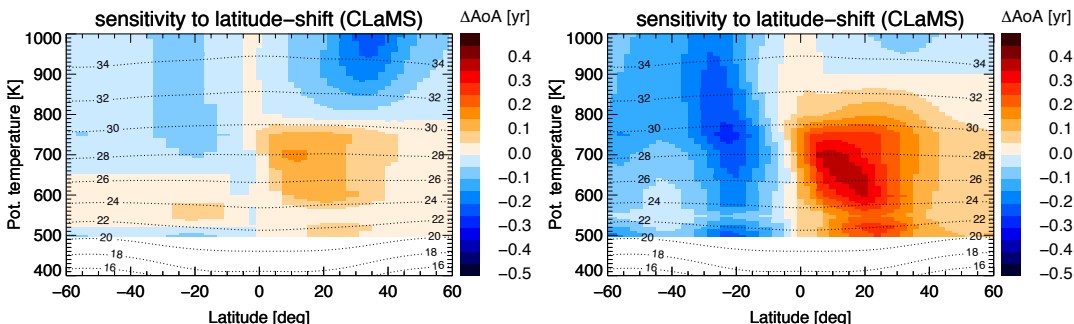

**Figure A4.** Left panel: Zonal mean AoA distribution of period I from CLaMS shifted by the shift profile plus 1-$\sigma$ uncertainty (reducing the shift below 800 K); right panel: same as left, but shifted by the shift profile minus 1-$\sigma$ uncertainty (increasing the shift below 800 K).

*Acknowledgements.* Part of this work was funded by the German Federal Ministry of Education and Research under grant no. 01LG1221B (ROMIC-BDChange) and under grant no. 01LG1222A (ROMIC-TRIP). For parts of the work, FP was funded by the Helmholtz Young Investigators group A-SPECi (Assessment of stratospheric processes and their effects on climate variability). We thank ECMWF for providing reanalysis data and the MLS team at Jet Propulsion Lab, Pasadena, CA, for providing MLS data. We acknowledge support by the Deutsche
5 Forschungsgemeinschaft and the Open Access Publishing Fund of the Karlsruhe Institute of Technology. The authors wish to thank two anonymous reviewers for their helpful and constructive comments.

The article processing charges for this open-access publication were covered by a Research Centre of the Helmholtz Association.

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
