# Peer review of "Shift of subtropical transport barriers explains observed hemispheric asymmetry of decadal trends of age of air"

_Atmospheric Chemistry and Physics, 2016_

## Referee Comment (RC1) · Anonymous Referee #1 · 12 Feb 2017

This paper uses satellite measurements and global model output to identify a recent shift in the latitudinal position of the stratospheric tropical pipe region and the subsequent impact on trend estimates throughout the stratosphere, especially in the extratropical regions where a north-south asymmetry has been found in the trends of mean age and a number of trace gases. This is an interesting analysis that highlights another complication in interpreting decadal scale trends and in comparing measurement and model trends.

My main suggestions are to clarify some of the figures and spend a little more time describing the techniques used. Some of the figures make it difficult to see the features described and the paper would benefit from adding a couple of more figures to

more clearly show the changes in the subtropical regions. The topic of the paper is appropriate for ACP so I recommend publication with consideration of the modifications suggested below.

Specific comments

Section 2.4: Need to include more description here, likely at least one equation, to help the reader understand the Miyazaki and Iwasaki method.

Figure 1: There is no color bar to indicate the mixing ratio values of the colors. Also, why are there gaps in the time series of black crosses, such as in the NH in 2009 and 2012? It would actually be nice to see the PDFs that you used to derive the transport barriers. This could include an average over a particular season for a few years, such as the 2005-8 period and the 2009-12 period. The NH and SH could be shown on the same plot to compare them. As it is, the color scale on Figure 1 makes it difficult to see how well the subtropical barrier represents the tracer gradient region.

Figure 3: Really hard to see everything in this plot. Too many lines and the filled contour colors are too similar to the over-plotted contours. It would also be nice to see a line plot for each of the two time periods of w* as a function of latitude, along with the transport barrier metrics, gradient genesis regions, etc. Might need to limit the number of lines you put on each plot though to make it clearer.

In Figures 2 and 3 the southern shift of the southern subtropical barrier is clear after 2009 but it should also be noted that it appears to move back north in 2014. This suggests the shift of the tropical pipe to the south may be a temporary one. I understand this is past the end of the MIPAS record and so doesn't affect the trend analysis. But it's still worth pointing out.

In Figures 4 and 5, the magnitudes of the changes explained by the shift, especially in the lower stratosphere in CLaMS, are lower than the total changes. This is mentioned in the text as perhaps due to a competing process or processes. Is it possible that your

shift of the tropical pipe in latitude is not enough at some levels? Is there a way to test how much shift can best explain the total changes?

Figure 6 shows a positive age trend everywhere but in the tropical lower stratosphere.

Minor comments

Pg. 7, line 1: change "for" to "of" Pg. 7, line 14: remove "in reality" Pg. 8, lines 2-3: "...found an increase in HCl volume mixing ratios in the Northern Hemisphere and a decrease in volume..." Pg. 8, line 4: "...change in age of air..." Pg. 10, line 15: "...during the westerly..."

---

## Referee Comment (RC2) · Anonymous Referee #2 · 17 Feb 2017

This study aims to demonstrate that a southward shift of the Brewer-Dobson circulation could be a reason for the observed spatial pattern of the age of air trend in the stratosphere during the MIPAS observation record. This is done by analyzing the changes in the subtropical transport barrier position and their impact on the age-of-air distribution using MIPAS observations and CLaMS model simulations driven by ERA-Interim. The obtained results can benefit the interpretation of the recent age-of-air trend and provide important implications regarding the changes in the Brewer-Dobson circulation. However, more careful descriptions of the methodology and results would be required. I would advise the authors to revise the manuscript accordingly. Below, I present my general remarks and specific points.

1. It is important to describe whether the overall summary and statistics are sensitive to the choice of the latitudinal shift. There are large interannual variations in the transport barrier position during the two periods, and the average period is short (i.e., four years). Also, the interannual variability is largely different between MIPAS and CLaMS. It is thus required to carefully describe the statistics of the latitudinal shift (e.g., the statistical significance of the trend at each levels) in MIPAS and CLaMS and their influences on the estimated impact on the age-of-air distribution.

2. The interannual variation of the transport barrier mostly disappeared after 2009 in both hemispheres in CLaMS and in the southern hemisphere in MIPAS (Fig. 2); this could explain large parts of the latitudinal shift between the two periods. The age spectrum at least should be obviously different in the absence of interannual variability, which may also influence the mean age through complicated transport processes, even if the period mean position is the same. This point needs to be discussed.

3. MLS data is used to evaluate the position of the transport barrier and is compared with MIPAS and CLaMS. Although the mean latitudinal shift is similar, there are large differences between MLS and MIPAS (and CLaMS), for instance, in 2008 in the northern hemisphere and in 2012 in the southern hemisphere. These differences need to be discussed more thoroughly, and summary statistics must be shown. Also, descriptions would be required on why both MLS and MIPAS are needed and why only MIPAS is used for the age-of-air calculation in this study. Information on the accuracy, precision, and coverage of each dataset would be helpful.

4. The CLaMS model performance needs to be evaluated more seriously. The authors show that the shift of the transport barrier position is similar between CLaMS and MIPAS. However, the mean position and the interannual variation exhibit large differences. Please provide a statistics summary on model performance and clarify if the model performance is sufficient for the purpose of this study.

5. It is described in P11L13 that the strongest negative trend of about -0.25

year/decade occurs in the northern tropics (from Fig. 6) and is consistent with trends derived from model calculations (e.g., Waugh, 2009), but this is confusing to me. The previous model calculations including the result of Waugh (2009) did not consider the effect of the latitudinal shift explicitly in their estimated age-of-air distribution, same as in the left panels in Fig 5 (not Fig. 6) in this study. I do not understand why these previous results can be compared with the result in this study after the influence of the latitudinal shift is removed (Fig. 6). I may be wrong, but further clarification would be useful.

- Specific comments:

P1L1" "is expected to accelerate..." Please describe what the expectation is based on.

P2L6: "380 and 420 K for the lower latitudes" Please describe the data used.

Section 2.3: Please describe the model resolution and discuss whether this is sufficient to realistically simulate the subtropical transport barrier.

Miyazaki and Iwasaki (2007) should be Miyazaki and Iwasaki (2008).

Figure 1: Color bars are required. Please change the color scale to clearly indicate the differences.

Figure 3: Please change the colors for the lines and shaded areas.

Figure 4: Please add the same results using MIPAS data and discuss the difference between CLaMS and MIPAS.

---

## Author Comment (AC1) · 28 Apr 2017

**Reply to Reviewer # 1**

We thank the reviewer for his/her insightful and constructive comments that definitely help to improve the paper. We have considered every comment carefully. Please find our replies below. The reviewer's comments are given in black, while our responses are in blue.

This paper uses satellite measurements and global model output to identify a recent shift in the latitudinal position of the stratospheric tropical pipe region and the subsequent impact on trend estimates throughout the stratosphere, especially in the extra-tropical regions where a north-south asymmetry has been found in the trends of mean age and a number of trace gases. This is an interesting analysis that highlights another complication in interpreting decadal scale trends and in comparing measurement and model trends.

My main suggestions are to clarify some of the figures and spend a little more time describing the techniques used. Some of the figures make it difficult to see the features described and the paper would benefit from adding a couple of more figures to more clearly show the changes in the subtropical regions. The topic of the paper is appropriate for ACP so I recommend publication with consideration of the modifications suggested below.

Specific comments

Section 2.4: Need to include more description here, likely at least one equation, to help the reader understand the Miyazaki and Iwasaki method.

This will be done. We will include the main equation by Miyazaki and Iwasaki, and provide some explanation. This will go into an appendix of the paper, together with the other explanations of methods requested.

Figure 1: There is no color bar to indicate the mixing ratio values of the colors.

We will improve this figure by providing a color bar and by using another color scale that makes it easier to see the variation of the N2O vmrs.

Also, why are there gaps in the time series of black crosses, such as in the NH in 2009 and 2012?

Gaps in the time series of black crosses appear where the pdf method to identify the position of the transport barrier was not successful. Due to the specific atmospheric situation the minimum in the N2O vmr pdf that marks the transport barrier can be such a broad and shallow valley that the determination of the absolute minimum fails or the uncertainty becomes very large. For these cases no latitudinal position of the transport barrier was derived. We will provide this information in the revised manuscript as well.

It would actually be nice to see the PDFs that you used to derive the transport barriers. This could include an average over a particular season for a few years, such as the 2005-8 period and the 2009-12 period. The NH and SH could be shown on the same plot to compare them. As it is, the color scale on Figure 1 makes it difficult to see how well the subtropical barrier represents the tracer gradient region.

We will provide in an appendix of the paper an example of the pdfs used to determine the latitudinal positions of the transport barriers, and explain along this example how the

method works. However, since the positions of the transport barriers have been derived on a monthly basis we will provide an example for a monthly average as well. The pdfs for the full periods would be so blurred due to the seasonal variations of the positions that they would not provide the required information. Further we will refer the paper by Palazzi et al. (2011) that presents examples of pdfs (their Fig. 3). As also requested by reviewer # 2, the color scale of Fig.1 will be changed so that the variation in the N2O vmrs can be seen more clearly.

Figure 3: Really hard to see everything in this plot. Too many lines and the filled contour colors are too similar to the over-plotted contours.

We will change the color table of the background so that it can be better distinguished from the colored lines.

It would also be nice to see a line plot for each of the two time periods of w* as a function of latitude, along with the transport barrier metrics, gradient genesis regions, etc. Might need to limit the number of lines you put on each plot though to make it clearer.

We will provide a line plot that contains the requested information for some example altitudes.

In Figures 2 and 3 the southern shift of the southern subtropical barrier is clear after 2009 but it should also be noted that it appears to move back north in 2014. This suggests the shift of the tropical pipe to the south may be a temporary one. I understand this is past the end of the MIPAS record and so doesn't affect the trend analysis. But it's still worth pointing out.

Yes, we'll do that (we do not think that the tropics will move to the South pole on the long term). We will make clear in the paper that it is natural variability on the time scale of (less than) a decade what we observe here.

In Figures 4 and 5, the magnitudes of the changes explained by the shift, especially in the lower stratosphere in CLaMS, are lower than the total changes. This is mentioned in the text as perhaps due to a competing process or processes. Is it possible that your shift of the tropical pipe in latitude is not enough at some levels? Is there a way to test how much shift can best explain the total changes?

In our paper we have applied the shift derived from the CLaMS data on distributions from CLaMS, and the shift derived from the observational data on the respective observational distributions, so the shifts and the tracer fields are treated in a self-consistent manner. In principle, we could find out by trial and error how far the distributions need to be shifted to explain most of the trend pattern. However, we do not see how this could bring us forward.

Figure 6 shows a positive age trend everywhere but in the tropical lower stratosphere.

The negative age trend extends to ±40° and up to 800 K (~ 30 km), that is more that the tropical lower stratosphere. Nevertheless, it is true that e.g. the age trend in the region shown by Engel et al. (2009) (mid-latitudinal mid-stratosphere) is indeed still positive. We will note this in the revised version of the paper.

Minor comments

Pg. 7, line 1: change "for" to "of" Pg. 7, line 14: remove "in reality" Pg. 8, lines 2- 3: ". . .found an increase in HCl volume mixing ratios in the Northern Hemisphere and a decrease in volume..." Pg. 8, line 4: "...change in age of air..." Pg. 10, line 15: ". . .during the westerly. . ."

Thank you for these corrections, we'll apply the changes.

---

## Author Comment (AC2) · 28 Apr 2017

**Reply to Reviewer # 2**

We thank the reviewer for his/her constructive and useful comments that help to improve the manuscript. We have considered every comment carefully. Please find our replies below. The reviewer's comments are in black, while our replies are in blue.

This study aims to demonstrate that a southward shift of the Brewer-Dobson circulation could be a reason for the observed spatial pattern of the age of air trend in the strato-sphere during the MIPAS observation record. This is done by analyzing the changes in the subtropical transport barrier position and their impact on the age-of-air distribution using MIPAS observations and CLaMS model simulations driven by ERA-Interim. The obtained results can benefit the interpretation of the recent age-of-air trend and provide important implications regarding the changes in the Brewer-Dobson circulation. However, more careful descriptions of the methodology and results would be required. I would advise the authors to revise the manuscript accordingly. Below, I present my general remarks and specific points.

1. It is important to describe whether the overall summary and statistics are sensitive to the choice of the latitudinal shift. There are large interannual variations in the transport barrier position during the two periods, and the average period is short (i.e., four years). Also, the interannual variability is largely different between MIPAS and CLaMS. It is thus required to carefully describe the statistics of the latitudinal shift (e.g., the statistical significance of the trend at each levels) in MIPAS and CLaMS and their influences on the estimated impact on the age-of-air distribution.

We will provide information on the uncertainties of the positions of the transport barriers, and on the uncertainty of the shift. These uncertainties will allow to judge if the shift is significant. Further, we will apply the (shift ± its uncertainty) on the age of air distributions of the first period and assess how far the resulting dipole patterns change within the shift uncertainties.

2. The interannual variation of the transport barrier mostly disappeared after 2009 in both hemispheres in CLaMS and in the southern hemisphere in MIPAS (Fig. 2); this could explain large parts of the latitudinal shift between the two periods.

It is true that the very strong QBO signal in CLaMS for the first period is no longer present in the second period. This might be in coincidence with the other changes happening in the stratosphere, i.e. another symptom of the same process, or an independent other process. We cannot judge from our observational basis which of the two possibilities apply. However, since we have selected the periods in line with the QBO oscillations we do not think that aliasing from the QBO variation leads to an extra shift between the periods.

The age spectrum at least should be obviously different in the absence of interannual variability, which may also influence the mean age through complicated transport processes, even if the period mean position is the same. This point needs to be discussed.

Within another study still to be published (Haenel et al., in preparation) we have applied altitude-dependent monthly mean zonal mean age of air (AoA) spectra calculated by CLaMS within the AoA calculation from MIPAS SF6 data and have analyzed the resulting AoA trends over the MIPAS observational period. The result, i.e. the altitude/latitude pattern of AoA trends was grossly the same as shown in this paper here. For this reason we do not think

that the changes in the age of air spectra have a major impact on the findings of our paper here. However, we will include this aspect in our discussion.

3. MLS data is used to evaluate the position of the transport barrier and is compared with MIPAS and CLaMS. Although the mean latitudinal shift is similar, there are large differences between MLS and MIPAS (and CLaMS), for instance, in 2008 in the northern hemisphere and in 2012 in the southern hemisphere. These differences need to be discussed more thoroughly, and summary statistics must be shown.

Fig.1 of this reply presents the correlation of transport barrier positions derived from MIPAS and MLS, respectively, for the overlapping mission period (2004 to 2012). The correlation is good and follows, except for some very few outliers, the 1:1 line. The length of the lines marking the crosses indicate the uncertainties of the derived barrier positions for MLS and MIPAS, respectively. The solid line in the panels is the pdf of the differences between positions derived from MIPAS and MLS, respectively. The mean over the pdf and its standard deviation is indicated in each panel. The differences of the positions have a very small bias (0.4 deg at most), and the standard deviation is by far smaller than the positions themselves, i.e. the derivation of positions from MIPAS and MLS is consistent. We consider this as a sufficient confirmation that MIPAS and MLS provides very similar information regarding the positions of the transport barriers and their variability. The figure and its discussion will be included into the appendix of the revised paper.

[Figure]

Fig. 1: Positions of transport barriers (crosses) from MIPAS N2O monthly averages (horizontal axis) vs. MLS N2O monthly averages (vertical axis) for the four seasons. The length of the lines marking the crosses indicate the uncertainty of the barrier positionderived from MIPAS and MLS, respectively. The solid line in the middle of the panel is the pdf of the differences of positions derived from MIPAS and MLS, respectively. Also provided in the panel is the mean over the pdfs and their standard deviations.

Also, descriptions would be required on why both MLS and MIPAS are needed and why only MIPAS is used for the age-of-air calculation in this study.

We have used both MIPAS and MLS data in order to demonstrate that two independent measurements provide consistent observations, and an instrument artifact can be largely ruled out (in case we used MIPAS only, both the AoA trend pattern and the shift of the transport barriers could be an instrumental artifact, for example a mis-location of the observations, occurring after a certain instant during mission lifetime). MIPAS is the only satellite instrument that provides age of air so far, as it is able to measure SF6 from which age of air is derived. This is the reason why only MIPAS data on age of air is used.

Information on the accuracy, precision, and coverage of each dataset would be helpful.

Some of this information has already been included in section 2 of the original paper. We will extend this section and provide more information on the data sets.

4. The CLaMS model performance needs to be evaluated more seriously. The authors show that the shift of the transport barrier position is similar between CLaMS and MIPAS. However, the mean position and the interannual variation exhibit large differences. Please provide a statistics summary on model performance and clarify if the model performance is sufficient for the purpose of this study.

The uncertainties of transport barrier positions and those of the shift will be added to Fig. 2 of the revised paper for CLaMS, too. The model performance with respect to the simulation of age of air and its variability has been analyzed in several papers, and good general agreement with observations has been found (see e.g. Pommrich et al., 2014; Ploeger et al., 2015). It is true that the transport barrier locations do not always agree between the three datasets regarding interannual variability, but the longer-term behavior (2002-2012) is very consistent, as is demonstrated in Fig. 2.

5. It is described in P11L13 that the strongest negative trend of about -0.25 year/decade occurs in the northern tropics (from Fig. 6) and is consistent with trends derived from model calculations (e.g., Waugh, 2009), but this is confusing to me. The previous model calculations including the result of Waugh (2009) did not consider the effect of the latitudinal shift explicitly in their estimated age-of-air distribution, same as in the left panels in Fig 5 (not Fig. 6) in this study. I do not understand why these previous results can be compared with the result in this study after the influence of the latitudinal shift is removed (Fig. 6). I may be wrong, but further clarification would be useful.

It is not expected that free-running climate models reproduce the actual short term variability well or, at least, that they generate this variability at the same time as it occurs naturally. By the way, this limitation is the main reasoning behind comparing "specified dynamics" model runs (i.e. model runs driven in some way by re-analyses) instead of free model runs to observational data records. A long-term climatological trend (30 years or more) is also expected to provide an average over the shorter-term variabilities.

In our case, we understand the overall observed variation (i.e. the observational "trend") to be caused by a long-term climatological part that can be captured by global climate models and, in addition, by some shorter-term (decadal or less) natural variability that is not well captured by the free-running climate models. We consider the shift of the circulation pattern as such a shorter-term variability. Removing that from the observational data should

leave us with the long-term climatological trend. The latter can be (and has been) compared with the climatological trend from climate models, and it has been demonstrated that the remaining trend is closer to the long-term climatological trend from the models. Further shorter-term variability besides the wobbling of the tropical pipe is possibly also present and has not been removed in our study. Therefore a perfect agreement between the "cleaned" observational trend and the climatological trend from models is not to be expected.

- Specific comments:

P1L1" "is expected to accelerate..." Please describe what the expectation is based on.

We have referred to the relevant literature in the first sentence of the introduction (p1, l11-13). Most of the climate models predict an acceleration of the Brewer-Dobson circulation as a consequence of global warming. This is also said in the introduction. We do not exactly understand what the comment refers to.

P2L6: "380 and 420 K for the lower latitudes" Please describe the data used.

We refer here to the paper by Ploeger et al. (2015). Data used within this paper were CLaMS model results and MIPAS observational AoA data, the same as used in this study here. We will add this information to the revised version of the paper.

Section 2.3: Please describe the model resolution and discuss whether this is sufficient to realistically simulate the subtropical transport barrier.

As CLaMS is a Lagrangian model, the grid is irregular and varying over time. Therefore it is only possible to state an average distance between the model air parcels as resolution. We will include this information into the text: "The model resolution of the CLaMS simulations considered here is about 100km in the horizontal direction. In the vertical direction, the resolution is about 400m around the tropical tropopause (see Pommrich et al., 2014 for details)."

Miyazaki and Iwasaki (2007) should be Miyazaki and Iwasaki (2008).

This will be corrected.

Figure 1: Color bars are required. Please change the color scale to clearly indicate the differences.

A color bar will be added, and the color scale of Fig.1 will be changed so that the variation in the N2O vmrs can be seen more clearly.

Figure 3: Please change the colors for the lines and shaded areas.

We will change the color table of the background so that it can be better distinguished from the colored lines.

Figure 4: Please add the same results using MIPAS data and discuss the difference between CLaMS and MIPAS.

The figure with data from CLaMS was added to demonstrate that the shift of the transport barrier, applied to the full surf zone area, reproduces the observed hemispheric dipole pattern, in other words, that not only the transport barrier, but the full surf zone area is shifted. We do not see what an additional figure from MIPAS would help here. MIPAS N2O data fully reproduce the finding from CLaMS. Eckert et al. (2014) (their Fig. 19) already demonstrated that a shift of the ozone distribution explains the observed ozone trends not only around the transport barrier, but in the surf zones as well. We consider these two cases to be sufficient confirmation that the observed hemispheric dipole pattern can be produced by such a shift of the low and mid-latitude distributions. In the revised paper, we'll refer to the ozone example in the Eckert et al. (2014) paper as a second example based on observational data.

---

## Author Response (AR1)

**Dear Editor:**

We thank the reviewers for their insightful and constructive comments that definitely helped to improve the paper. We have considered every comment carefully. Please find our replies and the related actions listed below. The reviewer's comments are given in black, while our responses and the related actions are in blue.

**Reply to Reviewer # 1**

**Reviewer comment:** This paper uses satellite measurements and global model output to identify a recent shift in the latitudinal position of the stratospheric tropical pipe region and the subsequent impact on trend estimates throughout the stratosphere, especially in the extra-tropical regions where a north-south asymmetry has been found in the trends of mean age and a number of trace gases. This is an interesting analysis that highlights another complication in interpreting decadal scale trends and in comparing measurement and model trends.

My main suggestions are to clarify some of the figures and spend a little more time describing the techniques used. Some of the figures make it difficult to see the features described and the paper would benefit from adding a couple of more figures to more clearly show the changes in the subtropical regions. The topic of the paper is appropriate for ACP so I recommend publication with consideration of the modifications suggested below.

Specific comments

Section 2.4: Need to include more description here, likely at least one equation, to help the reader understand the Miyazaki and Iwasaki method.

**Reply/Action:** This has been done. We have include an Appendix in the revised version that presents and discusses the main equation by Miyazaki and Iwasaki (2008), and provide some explanation.

**Reviewer comment:** Figure 1: There is no color bar to indicate the mixing ratio values of the colors.

**Reply/Action:** We have improved this figure by providing a color bar and by using another color scale that makes it easier to see the variation of the $N_2O$ vmrs.

**Reviewer comment:** Also, why are there gaps in the time series of black crosses, such as in the NH in 2009 and 2012?

**Reply:** Gaps in the time series of black crosses appear where the PDF method to identify the position of the transport barrier was not successful. Due to the specific atmospheric situation the minimum in the $N_2O$ vmr PDF that marks the transport barrier can be such a broad and shallow valley that the determination of the absolute minimum fails or the uncertainty becomes very large. For these cases no latitudinal position of the transport barrier was derived.

**Action:** We have now provided this information in the figure caption. We have included a section in a newly added Appendix where we describe the method in more detail. There we also explain why the method sometimes fails to determine a barrier position.

**Reviewer comment:** It would actually be nice to see the PDFs that you used to derive the transport barriers. This could include an average over a particular season for a few years, such as the 2005-8 period and the 2009-12 period. The NH and SH could be shown on the same plot to compare them. As it is, the color scale on Figure 1 makes it difficult to see how well the subtropical barrier represents the tracer gradient region.

**Reply/Action:** We have provided in an Appendix of the paper an example of the PDFs used to determine the latitudinal positions of the transport barriers, and explain along this example how the method works. However, since the positions of the transport barriers have been derived on a monthly basis we have provided an example for a monthly average as well. The PDFs for the full periods would be so blurred due to the seasonal variations of the positions that they would not provide the required information. As also requested by reviewer # 2, the color scale of Fig. 1 has been changed so that the variation in the $N_2O$ vmrs can be seen more clearly.

**Reviewer comment:** Figure 3: Really hard to see everything in this plot. Too many lines and the filled contour colors are too similar to the over-plotted contours.

**Reply/Action:** We have changed the color table of the background so that it can be better distinguished from the colored lines, and we have reduced the number of extra lines.

**Reviewer comment:** It would also be nice to see a line plot for each of the two time periods of w* as a function of latitude, along with the transport barrier metrics, gradient genesis regions, etc. Might need to limit the number of lines you put on each plot though to make it clearer.

**Reply/Action:** We have included an additional Figure (Fig. 4 in the revised manuscript) that contains the requested information for an example potential temperature level (600K).

**Reviewer comment:** In Figures 2 and 3 the southern shift of the southern subtropical barrier is clear after 2009 but it should also be noted that it appears to move back north in 2014. This suggests the shift of the tropical pipe to the south may be a temporary one. I understand this is past the end of the MIPAS record and so doesn't affect the trend analysis. But it's still worth pointing out.

**Reply/Action:** We have made clear in the paper that it is natural variability on the time scale of (less than) a decade what we observe here, and we mention explicitly that the tropical pipe moves back at the end of the MLS observation and CLaMS modeling period.

**Reviewer comment:** In Figures 4 and 5, the magnitudes of the changes explained by the shift, especially in the lower stratosphere in CLaMS, are lower than the total changes. This is mentioned in the text as perhaps due to a competing process or processes. Is it possible that your shift of the tropical pipe in latitude is not enough at some levels? Is there a way to test how much shift can best explain the total changes?

**Reply:** In our paper we have applied the shift derived from the CLaMS data on distributions from CLaMS, and the shift derived from the observational data on the respective observational distributions, so the shifts and the tracer fields are treated in a self-consistent manner. We demonstrate now in Appendix B that the change in the CLaMS age of air pattern can be almost reproduced quantitatively by a shift $(S - \Delta S)$ (where S is the derived

shift and ΔS is its 1-σ uncertainty), i.e. the change of the age pattern is within the uncertainty of the shift also for CLaMS.

**Action:** Discussion added in Appendix B.

**Reviewer comment:** Figure 6 shows a positive age trend everywhere but in the tropical lower stratosphere.

**Reply:** The negative age trend extends to ±40° and up to 800 K (~ 30 km), that is more that the tropical lower stratosphere. Nevertheless, it is true that e.g. the age trend in the region shown by Engel et al. (2009) (mid-latitudinal mid-stratosphere) is indeed still positive.

**Action:** We have noted this in the revised version of the paper.

**Reviewer comment:** Minor comments

Pg. 7, line 1: change "for" to "of" Pg. 7, line 14: remove "in reality" Pg. 8, lines 2- 3: ". . .found an increase in HCl volume mixing ratios in the Northern Hemisphere and a decrease in volume..." Pg. 8, line 4: "...change in age of air..." Pg. 10, line 15: ". . .during the westerly. . ."

**Reply/Action:** We have applied all suggested changes.

**Reply to Reviewer # 2**

**Reviewer comment:** This study aims to demonstrate that a southward shift of the Brewer-Dobson circulation could be a reason for the observed spatial pattern of the age of air trend in the stratosphere during the MIPAS observation record. This is done by analyzing the changes in the subtropical transport barrier position and their impact on the age-of-air distribution using MIPAS observations and CLaMS model simulations driven by ERA-Interim. The obtained results can benefit the interpretation of the recent age-of-air trend and provide important implications regarding the changes in the Brewer-Dobson circulation. However, more careful descriptions of the methodology and results would be required. I would advise the authors to revise the manuscript accordingly. Below, I present my general remarks and specific points.

1. It is important to describe whether the overall summary and statistics are sensitive to the choice of the latitudinal shift. There are large interannual variations in the transport barrier position during the two periods, and the average period is short (i.e., four years). Also, the interannual variability is largely different between MIPAS and CLaMS. It is thus required to carefully describe the statistics of the latitudinal shift (e.g., the statistical significance of the trend at each levels) in MIPAS and CLaMS and their influences on the estimated impact on the age-of-air distribution.

**Reply/Action:** We have provided information on the uncertainties of the positions of the transport barriers within an extra figure (Fig. A2 of the revised manuscript). Further we have added information on the uncertainty of the shift itself, by adding error ranges of the shift to Figure 2. We have applied the (shift ± its uncertainty) on the age of air distributions of the first period from CLaMS and have assessed how far the resulting dipole patterns change

within the shift uncertainties (Figure A3 of the revised manuscript). We have found that a (shift minus its uncertainty) can almost completely explain the changes modeled by ClaMS, i.e. the changes modeled by CLaMS lie within the uncertainty range of the shift. All extra figures mentioned here are part of the newly added Appendix.

**Reviewer comment:** 2. The interannual variation of the transport barrier mostly disappeared after 2009 in both hemispheres in CLaMS and in the southern hemisphere in MIPAS (Fig. 2); this could explain large parts of the latitudinal shift between the two periods.

**Reply:** It is true that the very strong QBO signal in CLaMS for the first period is no longer present in the second period. This might be coinciding with the other changes happening in the stratosphere, i.e. another symptom of the same process, or an independent other process. We cannot judge from our observational basis which of the two possibilities applies. However, since we have selected the periods in line with the QBO oscillations we do not think that aliasing from the QBO variation leads to an extra shift between the periods.

**Action:** We now mention that we have selected the periods such that the QBO oscillation is in the same phase at the beginning of the period; in the discussion, we have included a sentence stating that the variation of the barrier positions due to QBO changes dramatically between period I and II, however, that we cannot establish any cause-and-effect relationship.

**Reviewer comment:** The age spectrum at least should be obviously different in the absence of interannual variability, which may also influence the mean age through complicated transport processes, even if the period mean position is the same. This point needs to be discussed.

**Reply:** Within another study still to be published (Haenel et al., in preparation) we have applied altitude-dependent monthly mean zonal mean age of air (AoA) spectra calculated by CLaMS within the AoA calculation from MIPAS SF6 data and have analyzed the resulting AoA trends over the MIPAS observational period. The result, i.e. the altitude/latitude pattern of AoA trends, was grossly the same as shown in this paper here. For this reason we do not think that the changes in the age of air spectra have a major impact on the findings of our paper here.

**Action:** We have included a sentence on this aspect in our discussion.

**Reviewer comment:** 3. MLS data is used to evaluate the position of the transport barrier and is compared with MIPAS and CLaMS. Although the mean latitudinal shift is similar, there are large differences between MLS and MIPAS (and CLaMS), for instance, in 2008 in the northern hemisphere and in 2012 in the southern hemisphere. These differences need to be discussed more thoroughly, and summary statistics must be shown.

**Reply/Action:** Fig.1 of this reply (a similar figure is included in the Appendix of the revised manuscript, it is Fig. A2 there) presents the correlation of transport barrier positions derived from MIPAS and MLS, respectively, for the overlapping mission period (2004 to 2012). The correlation is good and follows, except for some very few outliers, the 1:1 line. The lengths of the cross bars indicate the uncertainties of the derived barrier positions for MLS and MIPAS, respectively. The solid line in the panels is the PDF of the differences between positions derived from MIPAS and MLS, respectively. The mean over the PDF and its standard deviation is indicated in the legend of each panel. The differences of the positions

have a very small bias (0.4° at most), and the standard deviation is by far smaller than the positions themselves, i.e. the derivation of positions from MIPAS and MLS is consistent. We consider this as a sufficient confirmation that MIPAS and MLS provide very similar information regarding the positions of the transport barriers and their variability. A slightly different figure and its discussion have been included into the appendix of the revised paper as Figure A2.

[Figure]

Fig. 1: Positions of transport barriers (crosses) from MIPAS $N_2O$ monthly averages (horizontal axis) vs. MLS $N_2O$ monthly averages (vertical axis) for the four seasons. The length of the cross bars indicates the uncertainty of the barrier position derived from MIPAS and MLS, respectively. The solid line in the middle of the panel is the PDF of the differences of positions derived from MIPAS and MLS, respectively. Also provided in the panel is the mean over the PDFs and their standard deviations.

**Reviewer comment:** Also, descriptions would be required on why both MLS and MIPAS are needed and why only MIPAS is used for the age-of-air calculation in this study.

**Reply**: We have used both MIPAS and MLS data in order to demonstrate that two independent measurements provide consistent observations, and an instrument artifact can be largely ruled out (in case we used MIPAS only, both the AoA trend pattern and the shift of the transport barriers could be an instrumental artifact, for example a mis-location of the observations, occurring after a certain instant during mission lifetime). MIPAS is the only satellite instrument that provides age of air so far, as it is able to measure SF6 from which age of air is derived. This is the reason why only MIPAS data on age of air is used.

**Action:** no action taken.

**Reviewer comment:** Information on the accuracy, precision, and coverage of each dataset would be helpful.

**Reply/Action:** Some of this information was already included in Section 2 of the original paper. We have extended this section and provide now more information on the data sets.

**Reviewer comment:** 4. The CLaMS model performance needs to be evaluated more seriously. The authors show that the shift of the transport barrier position is similar between CLaMS and MIPAS. However, the mean position and the interannual variation exhibit large differences. Please provide a statistics summary on model performance and clarify if the model performance is sufficient for the purpose of this study.

**Reply/Action:** The uncertainties of transport barrier positions and those of the shift have been added to Fig. 2 of the revised paper for CLaMS, too. We now point out that the model performance with respect to the simulation of age of air and its variability has been analyzed in several papers, and good general agreement with observations has been found (see e.g. Pommrich et al., 2014; Ploeger et al., 2015). In addition, new Figure 4 of the revised paper now demonstrates that behavior of the transport barrier locations averaged over the two periods is consistent with the observations, although they do not always agree between the three datasets regarding inter-annual variability.

**Reviewer comment:** 5. It is described in P11L13 that the strongest negative trend of about - 0.25 year/decade occurs in the northern tropics (from Fig. 6) and is consistent with trends derived from model calculations (e.g., Waugh, 2009), but this is confusing to me. The previous model calculations including the result of Waugh (2009) did not consider the effect of the latitudinal shift explicitly in their estimated age-of-air distribution, same as in the left panels in Fig 5 (not Fig. 6) in this study. I do not understand why these previous results can be compared with the result in this study after the influence of the latitudinal shift is removed (Fig. 6). I may be wrong, but further clarification would be useful.

**Reply:** It is not expected that free-running climate models reproduce the actual short term variability well or, at least, that they generate this variability at the same time as it occurs naturally. By the way, this limitation is the main reasoning behind comparing "specified dynamics" model runs (i.e. model runs driven in some way by re-analyses) instead of free model runs to observational data records. A long-term climatological trend (30 years or more) is also expected to provide an average over the shorter-term variabilities.

In our case, we understand the overall observed variation (i.e. the observational "trend") to be caused by a long-term climatological part that can be captured by global climate models and, in addition, by some shorter-term (decadal or less) natural variability that is not well captured by the free-running climate models. We consider the shift of the circulation pattern as such a shorter-term variability. Removing that from the observational data should leave us with the long-term climatological trend. The latter can be (and has been) compared with the climatological trend from climate models, and it has been demonstrated that the remaining trend is closer to the long-term climatological trend from the models. Further shorter-term variability besides the wobbling of the tropical pipe is possibly also present and has not been removed in our study. Therefore a perfect agreement between the "cleaned" observational trend and the climatological trend from models is not to be expected.

**Action:** no action taken

**Reviewer comment:** - Specific comments:

P1L1" "is expected to accelerate..." Please describe what the expectation is based on.

**Reply:** We have referred to the relevant literature in the first sentence of the introduction (p1, l11-13). Most of the climate models predict an acceleration of the Brewer-Dobson circulation as a consequence of global warming. This is also said in the introduction. We do not exactly understand what the comment refers to.

**Action:** no action taken.

**Reviewer comment:** P2L6: "380 and 420 K for the lower latitudes" Please describe the data used.

**Reply/Action:** We refer here to the paper by Ploeger et al. (2015). Data used within this paper were CLaMS model results and MIPAS observational AoA data, the same as used in this study here. We have added this information to the revised version of the paper.

**Reviewer comment:** Section 2.3: Please describe the model resolution and discuss whether this is sufficient to realistically simulate the subtropical transport barrier.

**Reply/Action**: As CLaMS is a Lagrangian model, the grid is irregular and varying over time. Therefore it is only possible to state an average distance between the model air parcels as resolution. We have included this information into the text: "The model resolution of the CLaMS simulations considered here is about 100 km in the horizontal direction. In the vertical direction, the resolution is about 400 m around the tropical tropopause (see Pommrich et al., 2014 for details)."

**Reviewer comment:** Miyazaki and Iwasaki (2007) should be Miyazaki and Iwasaki (2008).

**Reply/Action:** This has been corrected.

**Reviewer comment:** Figure 1: Color bars are required. Please change the color scale to clearly indicate the differences.

**Reply/Action:** A color bar has been added, and the color scale of Fig.1 has been changed so that the variation in the $N_2O$ vmrs can be seen more clearly.

**Reviewer comment:** Figure 3: Please change the colors for the lines and shaded areas.

**Reply/Action:** We have changed the color table of the background so that it can be better distinguished from the colored lines.

**Reviewer comment:** Figure 4: Please add the same results using MIPAS data and discuss the difference between CLaMS and MIPAS.

**Reply:** The figure with data from CLaMS was added to demonstrate that the shift of the transport barrier, applied to the full surf zone area, reproduces the observed hemispheric dipole pattern, in other words, that not only the transport barrier, but the full surf zone area is shifted. We do not see what an additional figure from MIPAS would help here. MIPAS $N_2O$ data fully reproduce the finding from CLaMS. Eckert et al. (2014) (their Fig. 19) already

demonstrated that a shift of the ozone distribution explains the observed ozone trends not only around the transport barrier, but in the surf zones as well. We consider these two cases to be sufficient confirmation that the observed hemispheric dipole pattern can be produced by such a shift of the low and mid-latitude distributions.

**Action:** In the revised paper, we refer to the ozone example in the Eckert et al. (2014) paper as a second example based on observational data.

[revised manuscript text omitted]

---

## Author Response (AR2)

**Dear Editor:**

Thank you for accepting our manuscript for final publication in ACP. Since the last revised version was accepted without changes, the submitted production files are identical in content with the last revised version.